# Microstructure and Properties of Nano-Hydroxyapatite Reinforced WE43 Alloy Fabricated by Friction Stir Processing

**DOI:** 10.3390/ma12182994

**Published:** 2019-09-16

**Authors:** Genghua Cao, Lu Zhang, Datong Zhang, Yixiong Liu, Jixiang Gao, Weihua Li, Zhenxing Zheng

**Affiliations:** 1School of Mechanical and Electronic Engineering, Guangdong Polytechnic Normal University, Guangzhou 510635, China; cghcaogenghua@126.com (G.C.); liuyixiong@gpnu.edu.cn (Y.L.); gjx205@163.com (J.G.); lwh927@163.com (W.L.); 2Guangdong Key Laboratory for Advanced Metallic Materials processing, South China University of Technology, Guangzhou 510641, China; lewzl@live.com (L.Z.); dtzhang@scut.edu.cn (D.Z.)

**Keywords:** WE43/HA composite, friction stir processing, microstructure, mechanical properties, corrosion behavior

## Abstract

This research mainly focuses on the successful fabrication of nano-hydroxyapatite (nHA) reinforced WE43 alloy by two-pass friction stir processing (FSP). Microstructure evolution, mechanical properties, and in vitro corrosion behavior of FSPed WE43/nHA composite and FSPed WE43 alloy were studied. The results show that nHA particles are effectively dispersed in the processing zone, and the well-dispersed nHA particles can enhance the grain refine effect of FSP. The average grain sizes of FSPed WE43 alloy and WE43/nHA composite are 5.7 and 3.3 μm, respectively. However, a slight deterioration in tensile strength and yield strength is observed on the WE43/nHA composite, compared to the FSPed WE43 alloy, which is attributed to the locally agglomerated nHA particles and the poor quality of interfacial bonding between nHA particles and matrix. The electrochemical test and in vitro immersion test results reveal that the corrosion resistance of the WE43 alloy is greatly improved after FSP. With the addition of nHA particles, the corrosion resistance of the WE43/nHA composite shows an even greater improvement.

## 1. Introduction

Magnesium and its alloys have several advantages when compared with traditional metal biomedical materials. The Young’s modulus and density of magnesium and its alloys are similar to that of natural bone, which can effectively avoid the stress shielding effect [1,2]. Magnesium is biodegradable in vivo, and the corrosion products have proven to be nontoxic. In addition, magnesium-based biomedical materials have been widely reported to positively stimulate the formation of new bone, which is favorable for bone fracture healing [3]. Therefore, magnesium alloys have great potential in applications as biodegradable metal materials [4,5]. However, biomedical magnesium alloys face the urgent issue of controlling corrosion behavior by avoiding local corrosion and controlling corrosion rates, in order to meet the safety and mechanical property requirements for biodegradable metal materials [6].

Hydroxyapatite (HA), the main inorganic component of human bone tissue and teeth, has emerged as a promising bioceramic material for its outstanding biocompatibility and bioactivity [7,8]. However, due to its brittleness and poor strength, HA in biomedical applications is currently limited to non-load bearing parts or low load-bearing parts. In this case, introducing HA particles into magnesium alloys is considered an effective method to improve the corrosion rate and biocompatibility of magnesium alloys. At present, several processes such as hot extrusion, stirred casting, powder metallurgy, and other methods are performed to prepare HA magnesium matrix composites with uniform corrosion behavior, good mechanical properties, and biocompatibility [9,10,11].

Friction stir processing (FSP) is an emerging solid-state processing technology for preparing fine-grained metal materials [12]. In recent years, due to its slight interface reaction between matrix and reinforced particles during processing, FSP has been used for metal matrix composite preparation [13,14,15]. For instance, nano-hydroxyapatite (nHA) particles have successfully been added to pure Mg substrate by multi-pass FSP, and the Mg/nHA composite shows preferable corrosion resistance in simulated body fluid (SBF) or Dulbecco’s phosphate buffered saline compared to the substrate. However, the investigations of mechanical properties are not mentioned in these papers [14,15]. As an implant material, the material should have reasonable mechanical properties in order to meet the clinical requirements of implantation, so investigation of these mechanical properties is also important. WE43 magnesium alloy with high strength and low cytotoxicity is suitable for biomedical applications. Therefore, in this research, the casted WE43 alloy was used as matrix, nano-sized HA particles as the reinforcing phase, and FSP was conducted to prepare WE43/nHA composites. The microstructure evolution during FSP, the effects of rotation speed on the distribution of nHA particles, and the effects of dispersed nHA particles on mechanical properties and corrosion behavior of the composites were studied.

## 2. Experimental Procedure

### 2.1. Raw Materials

Commercially available WE43 magnesium alloy (as-cast) sheets of size 150 mm × 30 mm × 6 mm (length × width × height) were used as the base metal (BM) in this study, and the chemical composition of the BM is shown in Table 1. Nano-sized HA powders (nHA) with a purity of >99% used in this study were purchased from Shanxi Baiwei Biotechnology Co., Ltd. (Xi’an, China). The TEM morphology of nHA particles is shown in Figure 1—the particles were of acicular morphology 20–30 nm in width and 60–120 nm in length.

### 2.2. Processing

Figure 2 presents the schematic illustration of the processing steps. To prepare the WE43/nHA composite, FSP was performed on BM sheets with a groove size of 1 mm × 4 mm (width × depth). After nHA particles were filled into the groove, a cylindrical and pin-less FSP tool was pressed down slowly until the shoulder contacted with the material, and it was then processed at a rotation speed of 600 rpm and a traverse speed of 60 mm/min along the groove direction. After processing, a metal sealing layer was formed above the groove, which could avoid the nHA particles escaping from the groove during FSP (Figure 2c). In the next processing step, cylindrical FSP tools with a shoulder diameter of 15 mm, consisting of a tapered cylindrical pin with a diameter varying from 2 to 5 mm over the length of 5 mm, were used to perform FSP on the sealed specimens.

A two-pass FSP with optimized processing parameters, including a rotation speed of 1000 rpm and a traverse speed of 60 mm/min, was employed to obtain WE43/nHA composites with a fine and uniform microstructure. The plunge depth of 0.8 mm and tilt angle of 2.5° relative to the normal direction of FSP plane were kept constant. The composite specimens obtained by FSP were coded as WE43/nHA. The same processing parameters and conditions were applied for conducting FSP on BM without the addition of nHA particles, with the obtained specimens being coded as FSP-WE43.

### 2.3. Characterization of Microstructure and Phase Composition

The specimens used for microstructure observation were mechanically grinded with emery papers (up to #5000 grade) and polished on a polishing machine. Further, the polished specimens were etched by picric acid solution (picric acid 5 g, alcohol 80 mL, acetic acid 10 mL, and deionized water 10 mL). Optical microscope (DM15000M, Leica, Wizlar, Germany) was used to observe microstructure at lower magnification. The distribution of second phases and reinforcements, as well as the fracture morphology, were observed by a scanning electron microscope (Nova Nano 430, FEI, Hillsboro, OR, USA). The morphology of the nHA particles was characterized by a transmission electron microscopy (JEM-2100F, JEOL, Tokyo, Japan).

### 2.4. Mechanical Properties Testing

The microhardness test was conducted on a HVS-1000 digital Vickers microhardness tester (YouHong, Corp., Shanghai, China) with the application of a load of 0.98 N and a loading cycle of 10 s. The indention interval was selected to be 0.5 mm in stirred zone (SZ), and every indentation was measured three times and the average value was calculated. Tensile specimens were machined parallel to the processing direction with the gauge being completely within the stirred zone, the shape and dimension of a tensile specimen is shown in Figure 3. The tensile test was carried out on a SANS CMT5105 universal tensile testing machine (MTS, Eden Prairie, MN, USA) with a strain rate of 2 × 10^−3^ s^−1^. At least five specimens were tested to evaluate the average property values.

### 2.5. Corrosion Behavior

#### 2.5.1. Electrochemical Test

Potentiodynamic polarization curve tests were performed on an electrochemical workstation (Vertex.5A. EIX, IVIUM, Eindhoven, the Netherlands) in SBF solution (8.035 g/L NaCl, 0.355 g/L NaHCO_3_, 0.225 g/L KCl, 0.231 g/L K_2_HPO_4_·3H_2_O, 0.311 g/L MgCl_2_·6H_2_O, 0.292 g/L CaCl_2_, 0.072 g/L Na_2_SO4, and 6.118 g/L Tris (HOCH2)_3_CNH_2_). The reference electrode was a saturated Ag/AgCl electrode and a platinum electrode was used as the counter electrode. One square cm area of the specimens was used as the working electrode. Specimens were exposed to the SBF solution for 30 min prior to the beginning of the experiments to establish open circuit potential. The potentiodynamic polarization was done between the potentials −2.5 and 0.5 V with a scanning rate of 5 mV/s.

#### 2.5.2. Immersion Test

The immersion test was performed per ASTM-G31-72 in SBF at 37 °C for 24, 48, and 72 h, respectively. Weight loss specimens of size 6 mm × 4 mm × 2 mm (length × width × height) were cut from the SZ of FSP-WE43 and WE43/nHA composite samples and BM as well, for the purpose of measuring the corrosion rate of specimens in SBF. After immersion, the corrosion products were removed by chromic acid (200 g/L CrO_3_, 10 g/L AgNO_3_, and 20 g/L Ba(NO_3_)_2_) and then ultrasonically cleaned in distilled water and ethanol, respectively. The weight of specimens was measured before and after immersion. The corrosion rate was calculated by the following equation:(1)CR=W/Atρ
where CR is the corrosion rate (mm/year); W represents the weight loss (g); A refers to the surface area (cm^2^); t is the immersion time; and ρ is the standard density of WE43. A density of 1.83 g/cm^3^ was used for all specimens in this study.

The corrosion morphology of specimens was observed by the SEM mentioned above.

## 3. Results and Discussion

### 3.1. Microstructure Evolution

Figure 4 shows the optical microstructure of BM and the stir zone of FSP-WE43 and WE43/nHA specimens. The average grain size of BM is measured ~50.9 μm. Grain refinement is achieved up to ~5.7 and ~3.3 μm in the FSP-WE43 specimen and WE43/nHA specimen, respectively. During FSP, materials in the stir zone will undergo dynamic recrystallization and coarse second phases will break into small particles, resulting from the severe plastic deformation caused by the FSP tool and thermal effect caused by friction [13,16]. This is the main reason for the apparent refinement of grains after FSP. During FSP of magnesium, the peak temperature of SZ is reported lower than 550 °C, at which temperature the nHA particles remain stable [17,18,19]. The incorporated insoluble nHA particles act to stimulate nucleation and impede the migration of grain boundaries [20,21]. As a result, further grain refinement is achieved on WE43/nHA composites.

### 3.2. Distribution of HA Particles

Figure 5 shows the distribution of the second phase particles and its corresponding EDS analysis in the WE43/nHA specimen. nHA particles are found well dispersed on the matrix after FSP, joined by only a few clusters with a diameter of less than 10 μm (Figure 5a). The high-angle annular dark field (HAADF) image and the corresponding EDS analyses of the stir zone (Figure 5b,c) confirm that nHA particles are successfully added into the WE43 alloy matrix and most nHA particles remain at nanoscale.

### 3.3. Mechanical Properties

#### 3.3.1. Microhardness

The microhardness distribution curves of FSP-WE43 and WE43/nHA specimens are plotted in Figure 6. After FSP, the Vickers microhardness value of the SZ is significantly increased in FSP-WE43 and WE43/nHA specimens. The microhardness value at the SZ of the FSP-WE43 and WE43/nHA specimens is relatively stable, while the microhardness value of the base metal region fluctuates. The region with a higher microhardness value is about 5 mm in width, which is approximately equal to the diameter of the pin on FSP tool. The mean microhardness value of WE43 substrate is ~62.6 HV, which increases to ~79.6 HV by FSP without the addition of nHA powder. In the case of introducing nHA powder during FSP, the mean microhardness value is improved up to ~85.2 HV.

The grain size of as-cast WE43 alloy is coarse and the microstructure is ununiform, so the microhardness value of the BM zone is lower and fluctuates. After FSP, the grains in the stir zone are remarkably refined and grain boundary strengthening is considered the main reason for the significant increase of microhardness value in the stir zone. For the WE43/nHA specimen, the distribution of microhardness values is related to the dispersion of nHA particles, and a relatively stable microhardness value fluctuation for the WE43/nHA specimen indicates that nHA particles are dispersed uniformly on WE43 substrate, which is consistent with the microstructure observation.

#### 3.3.2. Tensile Properties

Figure 7 shows a comparison of the tensile properties of the base metal and processed WE43 (with or without adding nHA particles) specimens. As plotted in Figure 7, the ultimate tensile strength (UTS), yield strength (YS), and elongation of specimens after processing are improved in different degrees compared with the BM. For BM, the YS and UTS are measured as only ~153.3 and ~193.2 MPa, respectively. After FSP, the YS and UTS of the FSP-WE43 specimen are improved up to ~198.7 MPa and ~255.4 MPa, respectively. Compared with the FSP-WE43 specimen, a slight decline in strength is observed on the WE43/nHA specimen, with the YS and UTS being measured as ~185.1 and ~232.3 MPa, respectively. Furthermore, the value of elongation after fracture is greatly increased by FSP as well. The elongation of BM is only ~5.2%, while in the FSP-WE43 and WE43/nHA specimen it is 20.2% and 10.1%, respectively.

#### 3.3.3. Fractographic Studies

Figure 8 shows the fractographic images of tensile specimens. The fractographic of BM (Figure 8a) consists of a large amount of cleavage facets and voids defects, which indicates that BM fails in a brittle way. Figure 8b shows the fractographic image of an FSP-WE43 specimen, in which a large amount of fine equal-sized equiaxed dimples can be observed. The fracture morphology exhibits plastic fracture characteristics. This confirms the high ductility of the FSP-WE43 specimen as shown in Figure 7. The fractographic images of the WE43/nHA specimen are represented in Figure 8c,d. Fine dimples can also be observed in the WE43/nHA specimen and partially agglomerated nHA particles can be seen (as shown by white circles). By observing agglomerated nHA particles at higher magnification, several cracks across the nHA cluster is found (Figure 8d). The loose nHA clusters can be the initiation source of cracks during failure, which is the main reason for the decline of elongation in the WE43/nHA specimen compared with that in the FSP-WE43 specimen.

In general, coarse grains and brittle Mg_12_Nd networks in as-cast WE43 magnesium alloy cause its poor tensile properties [22]. As shown in Figure 4, remarkable grain refinement is achieved and coarse Mg_12_Nd phases are broken into fine particles after FSP. Under the combined effects of grain boundary strengthening and dispersion strengthening, the tensile strength of processed specimens (with or without addition of nHA particles) are significantly improved. In addition, FSP eliminates the voids defects in as-cast WE43, which is beneficial to the improvement of strength and ductility. However, the localized nHA agglomerates reduce the tensile properties of the composites to a certain extent compared with those in the FSP-WE43 alloy, while the tensile properties of the WE43/nHA specimen are still improved compared with those in the BM.

### 3.4. Corrosion Behavior

#### 3.4.1. Electrochemical Test

The potentiodynamic polarization curves of as-cast WE43 and processed WE43 specimens are demonstrated in Figure 9. The corrosion potentials (E_corr_) and corrosion current density (i_corr_) of BM are measured as −1.691 mV (vs. Ag/AgCl) and 109.6 μA/cm^2^, respectively. For the processed specimens, the E_corr_ of the WE43/nHA specimen (−1.661 mV) shifts toward the positive side and the i_corr_ (46.7 μA/cm^2^) is lower than that of BM. The E_corr_ and i_corr_ values of the FSP-WE43 sample are measured as −1.678 mV and 53.7 μA/cm^2^, respectively. The highest E_corr_ value and lowest i_corr_ value indicate that the WE43/nHA composite has the best corrosion resistance in this study.

As discussed in Section 3.1, grain size is significantly refined by FSP. For magnesium alloy, fine grains are proved beneficial to the formation of a passive layer, as the result of increasing the number of grain boundaries per unit volume and reducing the galvanic couple between grain boundary and grain interior [23,24]. This is the main reason for the increase of corrosion resistance on processed WE43 specimens. Moreover, tiny dispersed nHA particles can also contribute to uniform corrosion behavior [25]. Therefore, the combined effects of grain refinement and dispersion of nHA particles lead to the improvement of corrosion resistance on WE43/nHA composites.

#### 3.4.2. Degradation in Immersion Test

Figure 10 shows the corrosion weight loss curve of BM, FSP-WE43, and WE43/nHA specimens. During the immersion period of 120 h, the corrosion weight loss of BM increases rapidly. After immersion for 120 h, BM specimens are almost completely degraded in the SBF solution. The corrosion weight loss rates of the FSP-WE43 specimen and WE43/nHA specimen are relatively stable. In the first 72 h, the weight loss rates of the FSP-WE43 and WE43/nHA specimen are about the same. After immersion for 72 h, the weight loss rate of the FSP-WE43 specimen increases rapidly, while that of the WE43/nHA specimen maintains a relatively stable value. The corrosion rates are calculated according to Equation (1). After immersion for 120 h, the corrosion rate of the BM specimen is 26.8 mm/year, and the corrosion rates of the FSP-WE43 and the WE43/nHA specimen are 8.1 mm/year and 3.9 mm/year, respectively. Obviously, the FSP process has greatly improved the corrosion resistance of the casted WE43 alloy, and the addition of nHA particles has further improved the corrosion resistance of the material.

Corrosion morphologies (with corrosion products) of specimens after immersion in SBF for 72 h are shown in Figure 11. The SEM images of corrosion morphology show that BM experiences severe localized corrosion after immersion for 72 h and the accumulation of thick corrosion products occur locally, which are reported to be Mg(OH)_2_ and calcium phosphate bio-minerals [26,27]. For the FSP-WE43 specimen, uniform protective films are observed in most areas, while a small number of protective films fall off locally, which may decrease the protective effect on corrosion attack. For the WE43/nHA specimen, a dense and uniform protective layer generates on the composite surface. These results indicate that the corrosion morphology is changed from local corrosion in as-cast WE43 alloy to uniform corrosion in FSP-WE43 alloy, which is attributed to the fine-grained and homogeneous microstructure by FSP. With the addition of dispersed nHA particles, the uniform corrosion morphology on WE43/nHA composite is more obvious.

Corrosion morphologies (without corrosion products) of specimens after immersion in SBF for 72 h are shown in Figure 12. It can be seen from Figure 12a that the BM specimen experiences extremely severe corrosion attack and a large amount of material is dissolved in SBF. Deep and large etch pits can be observed (Figure 12b), proving that the material has been eroded by SBF and the corrosion products cannot prevent further corrosion. However, the FSP-WE43 specimen still keeps a relatively complete surface morphology, although parts of materials are dissolved in SBF (Figure 12c,d). For the composite specimen, the original shape is almost maintained after immersion in SBF for three days and only shallow corrosion pits can be observed locally (Figure 12e,f). The corrosion morphology observation indicates that the corrosion resistance of the WE43/nHA composite is superior to that of the FSP-WE43 alloy and much superior to that of the as-cast WE43 alloy, which is in accord with the electrochemical test results.

## 4. Conclusions

Fine-grained WE43/nHA composite was successfully prepared through friction stir processing. Microstructure evolution and mechanical properties as well as in vitro corrosion behavior of WE43/nHA composite were studied. The main findings are summarized as follows:After friction stir processing, nHA particles disperse uniformly on WE43 matrix, and the dispersed nHA particles enhance the grain refinement effect during processing.The tensile properties of the WE43/nHA composite are significantly improved compared with those of the casted WE43 alloy, while experiencing a slight deterioration compared with the tensile properties of the FSP-WE43 alloy, which are the result of the locally agglomerated nHA particles and the poor quality of interfacial bonding between nHA particles and matrix.Due to the grain refinement and dispersed nHA particles, the corrosion resistance of the WE43/nHA composite is superior to that of the FSP-WE43 alloy and much superior to that of the as-cast WE43 alloy.

## Figures and Tables

**Figure 1 materials-12-02994-f001:**
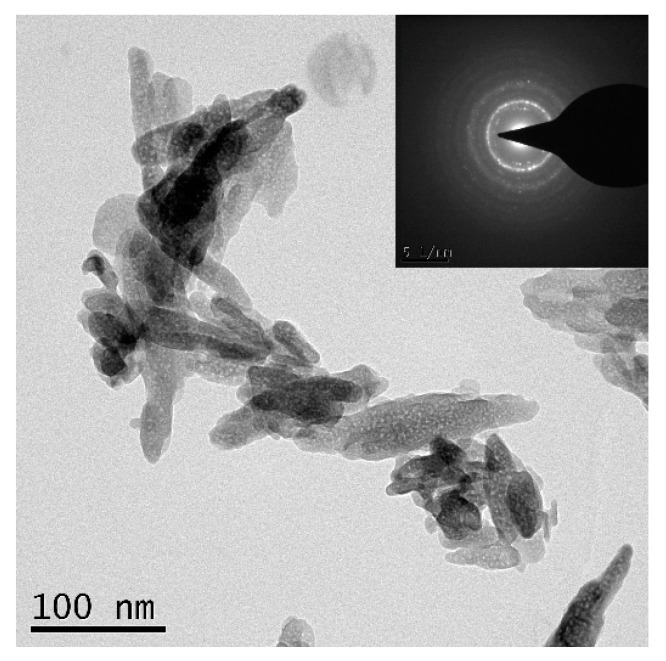
TEM morphology of nano-hydroxyapatite (nHA) particles.

**Figure 2 materials-12-02994-f002:**
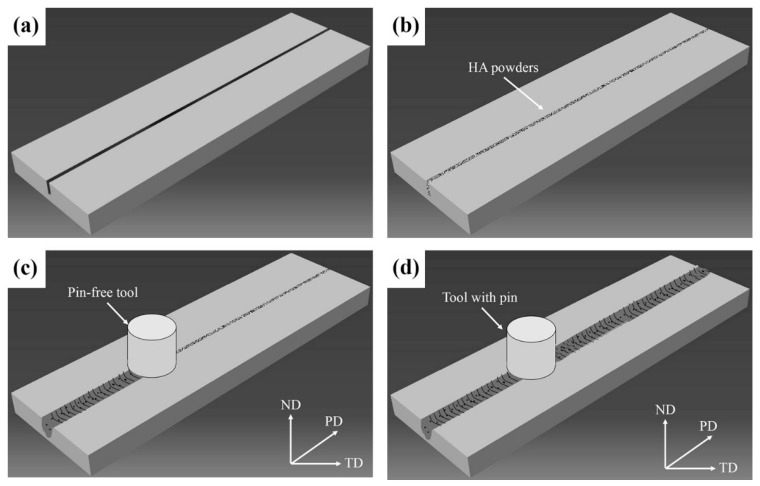
Schematic illustration of the processing steps. (**a**) Step 1; (**b**) Step 2; (**c**) Step 3; (**d**) Step 4.

**Figure 3 materials-12-02994-f003:**
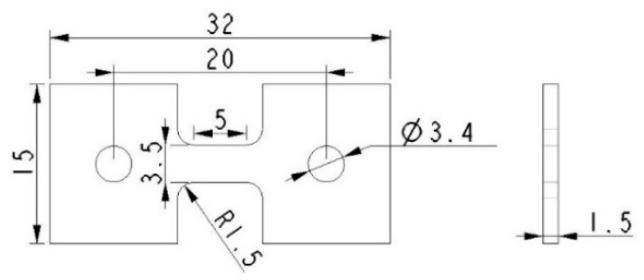
Shape and dimension of tensile specimen.

**Figure 4 materials-12-02994-f004:**
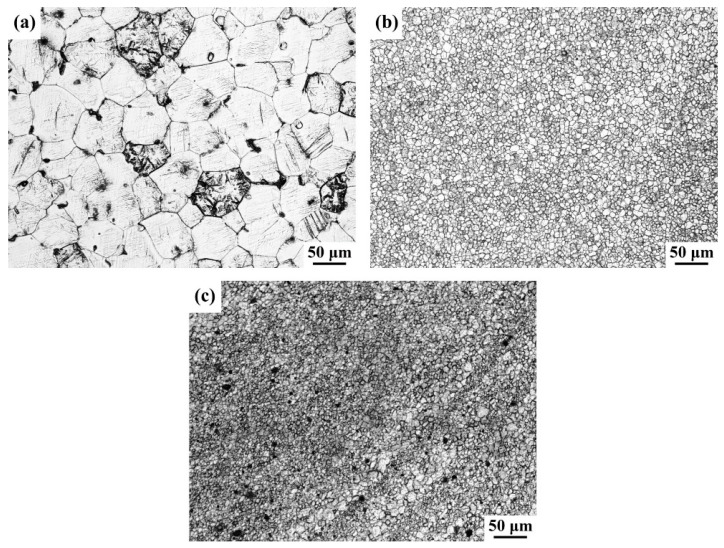
Optical image of (**a**) base metal (BM); (**b**) friction stir processing (FSP)-WE43; and (**c**) WE43/nHA specimens.

**Figure 5 materials-12-02994-f005:**
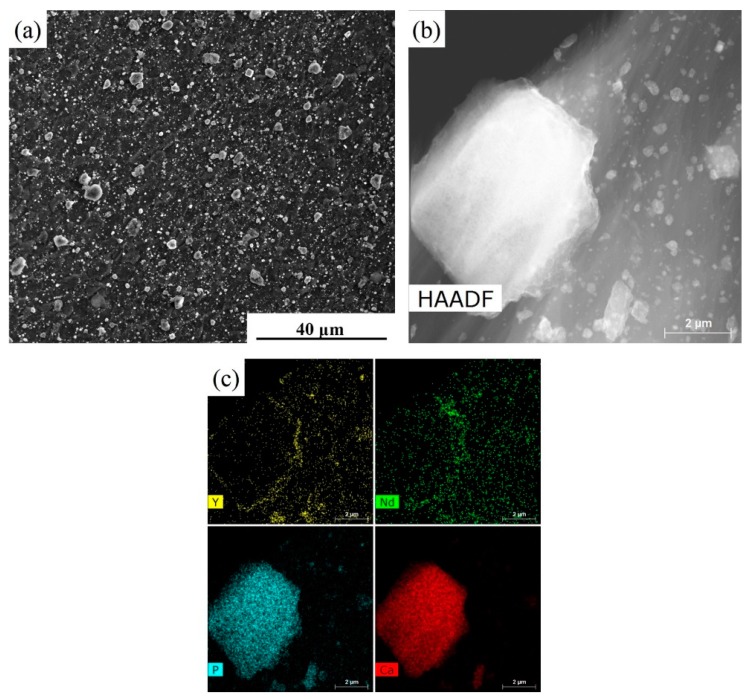
(**a**) SEM image; (**b**) TEM image; and (**c**) EDS analysis of the WE43/nHA specimen.

**Figure 6 materials-12-02994-f006:**
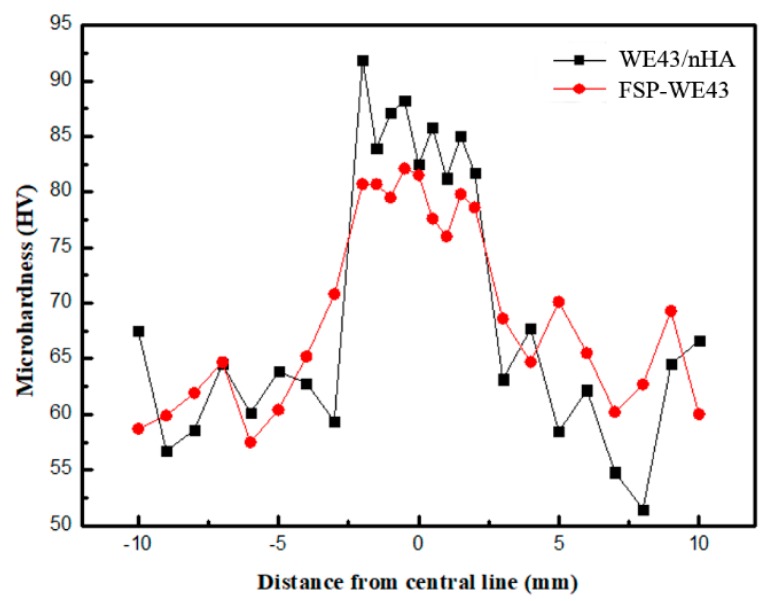
Microhardness distribution curves of WE43/nHA and FSP-WE43 specimens.

**Figure 7 materials-12-02994-f007:**
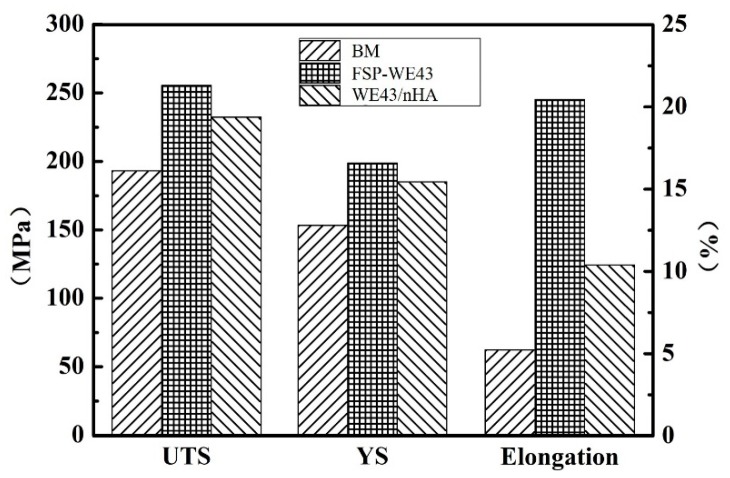
Tensile properties of BM, FSP-WE43, and WE43/nHA specimens. UTS: ultimate tensile strength; YS: yield strength.

**Figure 8 materials-12-02994-f008:**
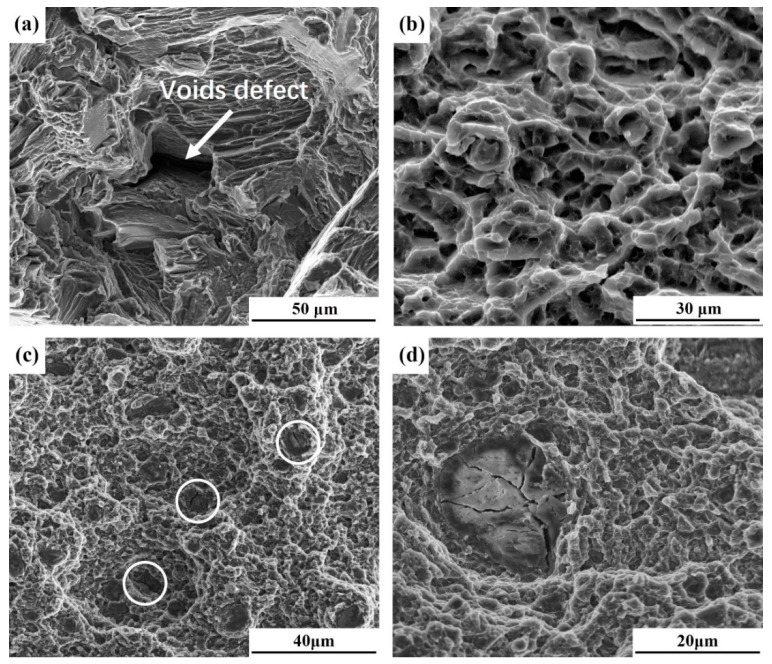
Fractographic images of (**a**) BM, (**b**) FSP-WE43, and (**c**) and (**d**) WE43/nHA specimens.

**Figure 9 materials-12-02994-f009:**
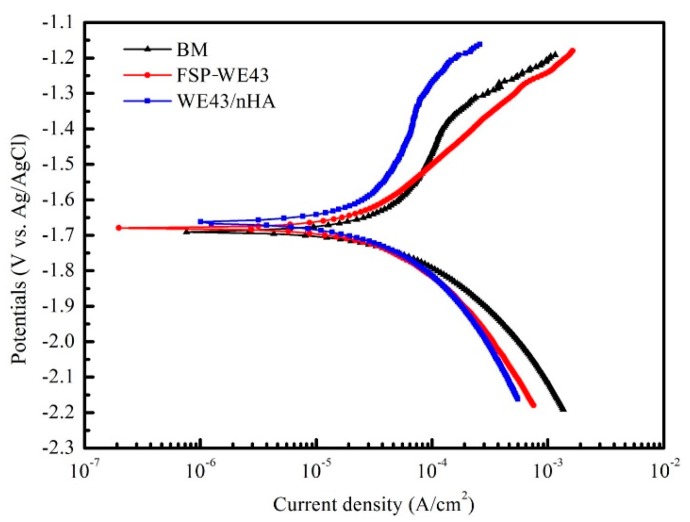
Potentiodynamic polarization curves of BM, FSP-WE43, and WE43/nHA specimens in simulated body fluid (SBF).

**Figure 10 materials-12-02994-f010:**
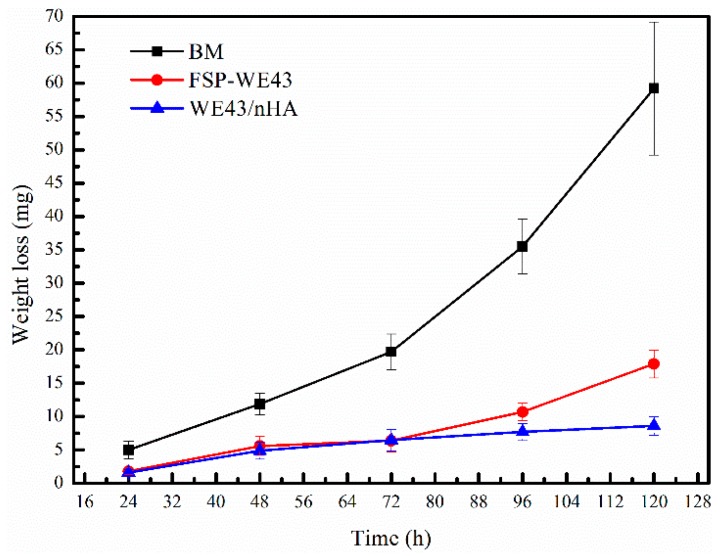
The weight loss and corrosion rate curves of immersion specimens.

**Figure 11 materials-12-02994-f011:**
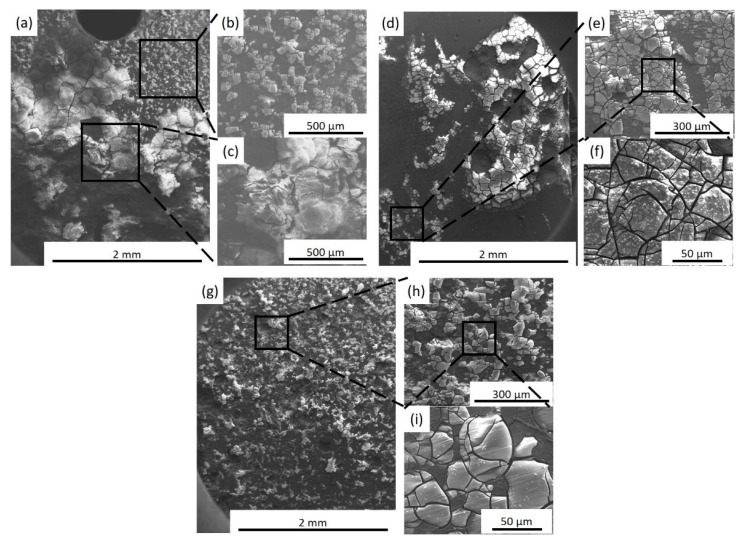
Corrosion morphologies of (**a**–**c**) BM, (**d**–**f**) FSP-WE43, and (**g**–**i**) WE43/nHA specimens after immersion in SBF for 72 h at 37 °C (with corrosion products).

**Figure 12 materials-12-02994-f012:**
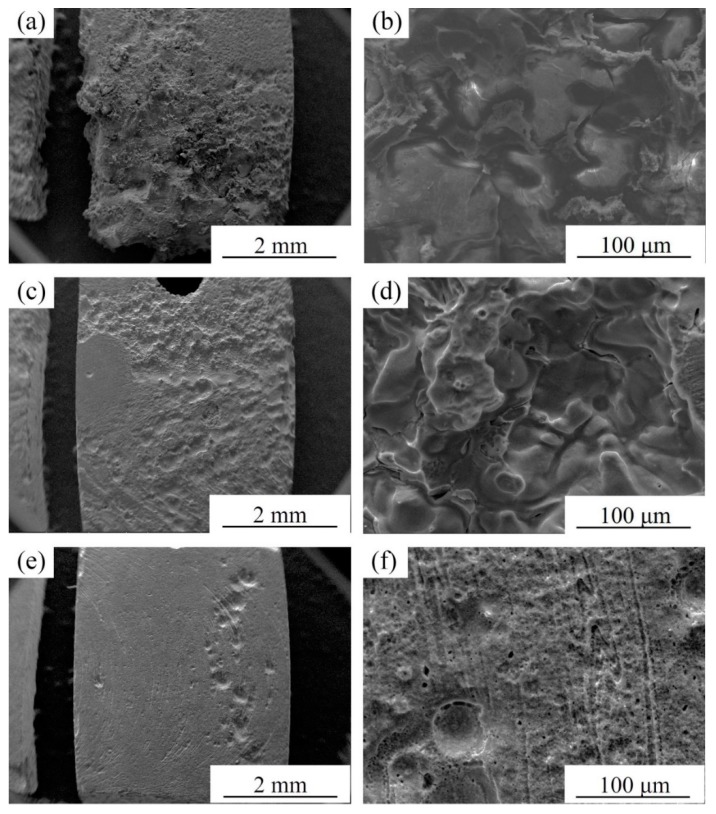
Corrosion morphologies of (**a**,**b**) BM, (**c**,**d**) FSP-WE43, and (**e**,**f**) WE43/nHA specimens after immersion in SBF for 72 h at 37 °C (without corrosion products).

**Table 1 materials-12-02994-t001:** Composites of as-cast base metal (wt. %).

Mg	Y	Nd	Gd	Zr	Ni	Ca	Mn	Si	Zn
Bal.	3.34	2.04	1.27	0.39	0.02	0.02	0.02	0.01	0.01

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
