# Peer review of "Microstructure and Properties of Nano-Hydroxyapatite Reinforced WE43 Alloy Fabricated by Friction Stir Processing"

_materials, 2019, doi:10.3390/ma12182994_

Round 1

Reviewer 1 Report

This paper examined the microstructure, the mechanical properties and the corrosion properties of a nHA particulate reinforced WE43 alloy produced by FSP. The study and the manuscript were well organized. The manuscript was orderly written. The reviewer had made the following comments:

1. Up to now, there have been several papers published regarding the addition of nHA particles in Mg-alloys or in Ti-alloys using FSP. For example, ref. 13and 14 cited in this manuscript and a paper published in Corrosion Science 104 (2016) 319–329 had disclosed many relevant information and knowledge regarding the processing, microstructure, mechanical and corrosion properties of the nHA reinforced magnesium alloys produced by FSP. I would not rate the novelty and the originality of this work as high since the mechanisms of strengthening and of corrosion protection reported in this work did not exceed those proposed by the paper of similar topics mentioned above except that WE43 as matrix material was not reported before.

2. The authors should give the range of the dimension of the nHA particles. In addition, it seems that the morphology and dimensions of the nHA in TEM image shown in fig. 1 did not agree with that of the n/HA stirred into WE43-matrix as revealed in fig 5(a) and (b). Fig. 5 shows substantial larger and equi-axed particles. Why?

3. The fractography of fig. 8 also reveals the HA particles of micron-size in dimension. The particles shown in fig.8(c) and (d) were not agglomerated particles. The morphology and dimensions of the nHA in fig.8 are also inconsistent with those in the TEM image in fig.1.

4. How a pin-less FSP-tool seal the groove requires more detailed description.

5. How were the tensile specimens sampled from the FSPed bead? Was the tensile direction along the FSP-bead or transverse to the bead?

6. The abbreviation SBF should be written out as "simulated body fluid".

7. The legends in fig. 7 standing for the materials are inconsistent with those used in the text and in previous figures.

8. Regarding the novelty and originality, the author should specify more specifically the contributions of this article compared to other published papers before this article could be published.

Author Response

Thank you very much for your review on our paper entitled "Microstructure and Properties of Nano-hydroxyapatite Reinforced WE43 Alloy Fabricated by Friction Stir Processing" (materials-559095). Based on your comments and requests, we have made some modifications on the original manuscript. Here below are the descriptions of revision.

1#Comments and Suggestions for Authors

This paper examined the microstructure, the mechanical properties and the corrosion properties of a nHA particulate reinforced WE43 alloy produced by FSP. The study and the manuscript were well organized. The manuscript was orderly written. The reviewer had made the following comments:

Up to now, there have been several papers published regarding the addition of nHA particles in Mg-alloys or in Ti-alloys using FSP. For example, ref. 13and 14 cited in this manuscript and a paper published in Corrosion Science 104 (2016) 319–329 had disclosed many relevant information and knowledge regarding the processing, microstructure, mechanical and corrosion properties of the nHA reinforced magnesium alloys produced by FSP. I would not rate the novelty and the originality of this work as high since the mechanisms of strengthening and of corrosion protection reported in this work did not exceed those proposed by the paper of similar topics mentioned above except that WE43 as matrix material was not reported before.

Answer: Great comment. As the reviewer said, Sunil (Mater. Sci. Eng. C 2014, 39, 315-324) (J Mater Sci: Mater Med (2014) 25:975–988) and Ahmadkhaniha (Corrosion Science 104 (2016) 319–329) systematically investigated the processing, microstructure and corrosion properties of the nHA reinforced pure Mg and AZ31 alloy produced by FSP, respectively. These papers provided effective guidance for our research, yet the investigations of mechanical properties are not mentioned in these papers. In general, the mechanical properties of WE43 is superior to pure Mg and AZ31 alloy, so it is more suitable for the application of bone implant. Therefore, we studied the mechanical properties of WE43/HA composites and the influence of HA particles on the fracture behavior of materials in this research.

The authors should give the range of the dimension of the nHA particles. In addition, it seems that the morphology and dimensions of the nHA in TEM image shown in fig. 1 did not agree with that of the n/HA stirred into WE43-matrix as revealed in fig 5(a) and (b). Fig. 5 shows substantial larger and equi-axed particles. Why?

Answer: Thanks for your suggestion. Figure 1 shows the original morphology of nHA particles, which were of acicular morphology with 20–30 nm in width and 60–120 nm in length. After FSP, HA particles agglomerated in the matrix to a certain extent, so the size increased slightly, and a few agglomerates reached the micron level.

The fractography of fig. 8 also reveals the HA particles of micron-size in dimension. The particles shown in fig.8(c) and (d) were not agglomerated particles. The morphology and dimensions of the nHA in fig.8 are also inconsistent with those in the TEM image in fig.1.

Answer: As mentioned above, a few agglomerated HA particles reached to the micron level after FSP, and we tested the composition of these particles with EDS, which showed that the particles were rich of Ca and P elements. It can be speculated that these agglomerated HA particles were mainly caused by the stirring breaking effect of stirring needle, leading to the result that they broke and reassembled. The fracture characteristics of loose HA clusters are shown in fig.8 (d).

How a pin-less FSP-tool seal the groove requires more detailed description.

Answer: Thanks for your suggestion. After HA particles were filled into the groove, a cylindrical and pin-less FSP tool was pressed down slowly until the shoulder contacted with the material, and then it was processed at a rotation speed of 600 rpm and a traverse speed of 60 mm/min along the groove direction. After processing, a metal sealing layer was formed above the groove, which could avoid the HA particles escaping from the groove during FSP (Fig. 2c).

How were the tensile specimens sampled from the FSPed bead? Was the tensile direction along the FSP-bead or transverse to the bead?

Answer: Tensile specimens were machined parallel to the processing direction with the gauge being completely within the stirred zone,

The abbreviation SBF should be written out as "simulated body fluid".

Answer: Thank you for pointing out the mistake. The "simulated body fluid" is used where it first appeared in the revised manuscript.

The legends in fig. 7 standing for the materials are inconsistent with those used in the text and in previous figures.

Answer: Thank you for pointing out the mistake. We modified it in the revised manuscript.

Regarding the novelty and originality, the author should specify more specifically the contributions of this article compared to other published papers before this article could be published.

Answer: Thank you for your suggestion and the helpful references. According to your suggestion and references, the differences between this article and other papers are added, please refer to the revised manuscript for detail (Page 2, lines 50-54).

The reviewer has helped us a lot in improving the quality of our manuscript. We would like to express our sincere thanks for the reviewer’s help.

Reviewer 2 Report

The manuscript under review presents the author's research on the   fabrication of nano-hydroxyapatite   reinforced WE43 alloy by two-passes of friction stir processing (FSP). Experimental results concerning microstructure, microhardness, tensile testing and electrochemical tests are provided. 

The paper has a good structure and the results seem interesting for the end users of this Mg alloy. However some points need further clarification:

Page 5, Fig. 4c: It is concluded from the metallographic image that he grain size is not uniform in the WE43/nHA specimens. Why??and How the authors have measured the grain size in this case?

Page 6, lines 153-154:The differences between the microhardness values reported for the WE43 and the WE43/nHA specimens specimens are really very low ( just 5HV) and despite the fact that this seems to be a systematic difference that is within the experimental error (or within the uncertainty of the measurement)

Page 6, lines 155-161: What is the size effect (grain size) on the microhardness measurements?

Page 6, lines 166-174: The uncertainty of the tensile test method (for the YS UTS and elongation) must be reported

Page 10, line268: "the pinning effect of dispersed HA"  There is no other reference throughout the text about the pinning effect of HA. This must be discussed and explained in a more detail manner.

The reviewer suggests major revision.

Author Response

Thank you very much for your review on our paper entitled "Microstructure and Properties of Nano-hydroxyapatite Reinforced WE43 Alloy Fabricated by Friction Stir Processing" (materials-559095). Based on your comments and requests, we have made some modifications on the original manuscript. Here below are the descriptions of revision.

2#Comments and Suggestions for Authors

The manuscript under review presents the author's research on the fabrication of nano-hydroxyapatite reinforced WE43 alloy by two-passes of friction stir processing (FSP). Experimental results concerning microstructure, microhardness, tensile testing and electrochemical tests are provided.

The paper has a good structure and the results seem interesting for the end users of this Mg alloy. However some points need further clarification:

Page 5, Fig. 4c: It is concluded from the metallographic image that he grain size is not uniform in the WE43/nHA specimens. Why??and How the authors have measured the grain size in this case?

Answer: The uneven size of WE43/nHA specimens is caused by the insufficient dispersion of HA particles during FSP, and the grain size in most areas is relatively uniform. Combined with a large number of OM images and SEM images, the average grain size was measured by the mean linear intercept method.

Page 6, lines 153-154:The differences between the microhardness values reported for the WE43 and the WE43/nHA specimens specimens are really very low ( just 5HV) and despite the fact that this seems to be a systematic difference that is within the experimental error (or within the uncertainty of the measurement)

Answer: In order to avoid experimental errors as much as possible, the indention interval was selected to be 0.5 mm in stirred zone, and every indentation was measured three times and the average value was calculated. According to the hardness curves, the microhardness of the WE43/nHA sample is higher than that of the FSP-WE43 sample, although the average hardness of the two samples is close to each other.

Page 6, lines 155-161: What is the size effect (grain size) on the microhardness measurements?

Answer: The same samples were used in the microhardness test and microstructure observation. The average grain size of BM,FSP-WE43 and WE43/nHA sample were about 50.9 μm, 5.7 μm and 3.3 μm, respectively.

Page 6, lines 166-174: The uncertainty of the tensile test method (for the YS UTS and elongation) must be reported

Answer: Thanks for your suggestion. In this research, at least five specimens were tested to evaluate the average property values in this research. We added it in the revised manuscript.

Page 10, line268: "the pinning effect of dispersed HA" There is no other reference throughout the text about the pinning effect of HA. This must be discussed and explained in a more detail manner.

Answer: Thanks for your suggestion. In our revised manuscript (Page 4, lines 139-141), we thought that “During FSP of magnesium, the peak temperature of SZ is reported lower than 550 ℃, at which the HA particles remain stable [17-19]. The incorporated insoluble HA particles act as stimulating nucleation and impede the migration of grain boundaries [20, 21].” However, we did not do more work to prove the pinning effect of dispersed HA, so we modified it to “the dispersed HA particles enhances the grain refinement effect during processing” in the revised manuscript.

The reviewer suggests major revision.

The reviewer has helped us a lot in improving the quality of our manuscript. We would like to express our sincere thanks for the reviewer’s help.

Round 2

Reviewer 1 Report

The authors had considered the reviewer's comments and have made several amendments accordingly. In reply to the comment 1, they had justified the contribution of this article by pointing out the superiority of the WE43 to Mg-alloys. The approx. dimensions of the nHA particles have been given. The seemingly inconsistency between fig.1 and fig.5 has been explained. The process regarding sealing the groove by a pinless tool has been described. The legends in fig.7 have been corrected.

I am delighted to recommend this revised article for publication in Materials.

Author Response

You has helped us a lot in improving the quality of our manuscript. We would like to express our sincere thanks for your help!

Reviewer 2 Report

The revised manuscript has been improved. The authors have taken into account all suggestions and addressed all open issues accordingly.

I recommend acceptance in the current form.

Author Response

You helped us a lot in improving the quality of our manuscript. We would like to express our sincere thanks for your help!